# Effects of Contamination by Heavy Metals and Metalloids on Chromosomes, Serum Biochemistry and Histopathology of the Bonylip Barb Fish Near Sepon Gold-Copper Mine, Lao PDR

**DOI:** 10.3390/ijerph17249492

**Published:** 2020-12-18

**Authors:** Latsamy Soulivongsa, Bundit Tengjaroenkul, Lamyai Neeratanaphan

**Affiliations:** 1Ecotoxicology, Natural Resources and Environment Project, Khon Kaen University, Khon Kaen 40002, Thailand; soulivongmee@gmail.com (L.S.); btengjar@kku.ac.th (B.T.); 2Division of Veterinary Medicine, Faculty of Veterinary Medicine, Khon Kaen University, Khon Kaen 40002, Thailand; 3Division of Environmental Science, Faculty of Science, Khon Kaen University, Khon Kaen 40002, Thailand

**Keywords:** cytogenetic, environment, fish, metal, pathology, serum enzyme

## Abstract

The objectives of the present study were to determine the concentrations of heavy metals and metalloids in water, sediment and *Osteochilus vittatus* fish, and to assess chromosome aberrations, serum biochemical changes and histopathological alterations in *O. vittatus* from the Nam Kok river near the Sepon gold-copper mine, Lao People’s Democratic Republic compared with the reference area. The results showed that Fe, Mn and Ni in water, As and Cd in sediment as well as As, Cd, Cr, Mn and Ni in *O. vittatus* muscle samples near the gold-copper mine exceeded standard values. Furthermore, the chromosome assessment in *O. vittatus* revealed seven types of chromosome aberrations, and the highest total number of chromosome aberrations was a centromere gap. The total number of chromosome aberrations, cell number with chromosome aberrations and percentage of chromosome aberrations in *O. vittatus* as well as serum liver enzymes between the studied areas were significantly different (*p* < 0.05). The liver histopathological alterations of the fish near the gold-copper mine revealed atypical cellular structures as nuclear membrane degeneration, rough endoplasmic reticulum disintegration and abnormal cytoplasmic mitochondria. The results of this study suggested that heavy metal and metalloid contaminations from the Sepon gold-copper mine area negatively affect *O. vittatus* fish in terms of chromosomal defects, serum biochemical changes and liver histopathological appearances.

## 1. Introduction

Laos has abundant mountainous landscapes with many important mineral resources, and presently the Lao government promotes mineral resources as the engine to develop the country through concessions to foreign companies [1,2]. Since 2007, production of mineral commodities included coal, gemstone, copper, zinc, silver, gold, barite and limestone, and in 2018, the key valuable minerals were copper with a value of USD 502 million followed by gold with a value of USD 200 million [3,4]. By 2020, Lao mining production will account for 30% of total metal export value, and contribute approximately 10% to Lao gross domestic product. The two largest gold mines in Laos accounting for 90% of country’s mining production were Phu Kham gold mine in Vientiane Province and Sepon gold-copper mine in Savannakhet Province. The Lao gold export is mainly to neighbouring countries such as Thailand, Vietnam, China and Malaysia [1,3]. Besides mining industry, fishery and aquaculture are other main important economic sectors in Lao People’s Democratic Republic and contribute to more than half of the animal protein source for Laotians [5]. The Lao government endeavours to grow annual fish consumption per capita; however, in some polluted area [6,7] the amount of fish captured from rivers and reservoirs in Lao People’s Democratic Republic is decreasing due to several aquatic environmental incidences, including the fish that died in a great amount along the river near the Lao Sepon gold-copper mine. Generally, metals and metalloids can leach out from the ill-managed gold mine processes to nearby aquatic ecosystems, and consequently, can cause detrimental effects to the aquatic creatures and human health due to their potential toxicities, persistence and tendency to bioaccumulate [8,9]. In the biochemical aspect, Fe, Zn, Co, Cu and Mn are classified as essential metal elements that have poisonous potentials when they are higher than the safe levels [10,11]. As, Cd, Cr, Pb and Ni are classified as non-essential metal elements that can cause toxicities as they are at trace levels. Several research studies mentioned that metals can negatively modify structural, physiological and biochemical consequences of animal and human organs. They could also cause genotoxicity, mutation and cancer [12,13,14,15,16].

Sediment and fish tissues are the main deposits of heavy metals and metalloids from electronic, industrial and mining areas more than in the water medium [17,18]. Measurements of metal and metalloid concentrations only in sediment or water do not provide sufficient information on the risk posed by metal and metalloid bioaccumulations or biomagnifications as in aquatic creatures [19]. To date, fish species are considered to be more suitable as bio-monitors for the assessment on effects of toxic elements in aquatic ecosystems [20] because they occupy a wide range of habitats and trophic levels with a relatively long life span and they are capable of absorbing and concentrating heavy metals and metalloids in their body tissues [21,22,23,24,25]. 

*O. vittatus* is an important fish species in Laos like other countries in Southeast Asia. It feeds on the roots of plants, algae and some crustaceans according to Shibukawa et al. [26]. From the previous information, the rapid growth of gold mining in Lao People’s Democratic Republic has potentially and intensely raised heavy metals and metalloid emissions into air, soil and water of aquatic ecosystems; however, data involving types and concentrations of metals and metalloid elements in environment and the health effects of these elements on *O. vittatus* are very limited. Therefore, a first time study on concentrations of As, Cd, Cr, Fe, Mn, Ni and Pb in the surrounding environment and their negative effects on chromosome aberrations, serum biochemical indices and liver cell pathology of *O. vittatus* in the Namkok river was conducted. The metals and metalloid impact information on several aspects of *O. vittatus* can be applied for environment and human health risk management as well as making guidelines, standards, regulations, legislations and policies for mining industry in Lao People’s Democratic Republic.

## 2. Materials and Methods

### 2.1. Study Area

The study area located at the Namkok river near a Sepon gold-copper mine, Viraboury District, Savannakhet Province, Lao People’s Democratic Republic, at GPS location 16°55′34.37″ N 105°59′58.16″ E, and the reference area without surrounding industrial, mining and electronic activities located at the Nam Souang river in Naxaythong District, Vientiane Capital, at GPS location 18°14′52.88″ N 102°27′54.78″ E (Figure 1).

### 2.2. Water Quality Parameters

Water quality parameters including dissolved oxygen (DO), potential of hydrogen (pH), temperature, total hardness (TH), carbonate hardness (CH) and electrical conductivity (EC) were measured at the experimental areas using mobile digital meters at 9:00 a.m., whereas nitrite, nitrate and ammonia were measured using the titration methods (Table 1).

### 2.3. Sample Collection

Three replicates of sediment, water and *O. vittatus* muscle samples were randomly collected from two sampling areas as the Nam Kok river near the Sepon gold-copper mine (study area) and the Nam Souang river (reference area or non-contaminated area) during July–October in 2019 (Figure 1). The water samples were fixed with nitric acid, and the sediment samples were air dried before quantifications of 7 heavy metals and a metalloid concentration. Ten *O. vittatus* samples were collected to investigate as follows: the heavy metals and metalloid accumulated in the muscles, chromosome aberrations from kidney cells, serum biochemistry and liver cell morphology. Liver cell structure changes were investigated by compare the liver cells between the fish from the reference and the study areas. Average body weights of *O. vittatus* samples taken from the study and the reference areas were approximately 123.00 ± 18.20 and 108.00 ± 20.74 g, respectively.

### 2.4. Measurements of the Heavy Metal and Metalloid Concentrations 

#### 2.4.1. Heavy Metal and Metalloid Measurements in Water Samples

A total of 25 mL of each water sample was added with 1.25 mL 30% HNO_3_, and then the sample was digested on a hot plate at 95 ± 5 °C for 1 h. After cooling, the digested sample was adjusted with volume to 25 mL with deionized water (DI), and filtered through polycellulose paper No.1. The final sample was analyses by Inductively Coupled Plasma Optical Emission Spectrometry (ICP-OES) [27].

#### 2.4.2. Heavy Metal and Metalloid Measurements in Sediment Samples

Exactly 1.0 g of each sediment sample was sequentially digested with 5 mL of nitric acid, 15 mL of hydrochloric acid and 10 mL of hydrogen peroxide in heating mantle to 95 ± 5 °C for 2 h. After cooling, the sample was adjusted volume to 50 mL with deionized water (DI) and filtered through filter paper No.42. The final samples were analysed by ICP-OES [27].

#### 2.4.3. Heavy Metal and Metalloid Measurements in *O. vittatus*

Exactly 1.0 g of each fish muscle was mixed with sulphuric and nitric acid and digested on a hot plate at 60 °C for 30 min. Then, 10 mL of hydrogen peroxide was added and digested on a hot plate at 95 ± 5 °C for 1 h. After cooling, the digested sample was adjusted volume to 25 mL with deionized water. Then, the mixture was passed through filtered paper No.1. The final sample was analysed by ICP-OES [27].

#### 2.4.4. Quality Control and Quality Assurance

The ICP-OES wavelength analyses for As, Cd, Cr, Pb, Ni, Mn and Fe were set at 188.979, 226.502, 267.716, 220.353, 231.604, 259.327 and 259.939 nm, respectively. Detection limits of analysed elements were 0.006 mg/kg for As; 0.001 mg/kg for Cd, Cr, Ni; 0.005 mg/kg for Pb; and 0.002 mg/kg for Mn, Fe. Analyses of blanks and standards were conducted at every 10th sample. The concentrations of the heavy metals and metalloid in the procedural blanks were significantly less than 5% of the mean analysed concentrations for all metals and metalloid. Replications of the analyses were conducted to guarantee the precision and accuracy of all measurements. The results were found not to deviate by more than 2% from the certified levels. The heavy metal and metalloid recovery values were calculated by acceptance in the range of 85–115%. The results at 90–100% of the acceptable values were considered as accurate [27,28].

### 2.5. Chromosome Preparation and Assessment

*O. vittatus* were injected with 0.05% colchicine and left for 1 h. Then, each fish kidney was cut into small pieces, mixed with 8 mL of 0.075 KCl for 25 min and centrifuged at 1500 rpm for 10 min. The kidney cells were fixed in a fresh and cool fixative (ratio 3 methanol:1 glacial acetic acid). The fixative was gradually added up to 8 mL before centrifuge again at 1500 rpm for 10 min, and then the supernatant was discarded. The fixation process was repeated 4 times until the supernatant was cleared. The sediment was mixed with 1 mL of fixative, and the cell suspension was dropped onto a glass slide. The air dried slide was conventionally stained with 20% Giemsa’s solution for 30 min [29,30]. Total observable and clear metaphase chromosomes of 500 *O. vittatus* kidney cells were photographed under a light microscope at magnification of 1000×. Total number of chromosome aberrations, number of cell with chromosome aberrations and percentage of chromosome aberrations were calculated and analysed statistically.

### 2.6. Serum Biochemistry Study

Fish blood samples were collected from the caudal vessel. Clotted blood samples were centrifuged at 2000× *g* for 10 min, and serum samples were stored at −20 °C prior to further biochemical analyses [31]. Serum total protein (TP), glucose (Glu), enzyme aspartate aminotransferase (AST) and alanine aminotransferase (ALT) were measured using Automate Analyzer (ROCHE/Hitachi Cobas C501, ROCHE/Hitachi, Roche Diagnostics K.K., Tokyo, Japan).

### 2.7. Histopathology Study

Liver samples were fixed in 1% osmium tetroxide for 1 day, dehydrated in series of ethyl alcohol, transferred to propylene oxide and embedded in capsule with eppon polymer. Ultrathin sections were stained with uranyl acetate and lead citrate and observed under a transmission electron microscope (JEOL 100, JEOL, JEOL Ltd., Tokyo, Japan) [32].

### 2.8. Statistical Analyses

Heavy metal and metalloid concentrations in contaminated water, sediment and *O. vittatus* muscle samples as well as serum biochemical levels were presented and tested with independent T-test. Total number of chromosome aberrations, cell number with chromosome aberrations and percentage of chromosome aberrations in the fish near the Sepon gold-copper mine and the reference areas were statistically analysed using Mann–Whitney U-test under the SPSS program version 24, at the 95% confidence level. Histological changes of the fish liver cells were reported descriptively.

## 3. Results

### 3.1. Water Quality Parameters

The values of water quality parameters from the study and the reference areas are shown in Table 2.

### 3.2. Heavy Metal and Metalloid Concentrations in Water Samples

The average concentrations of As, Cd, Cr, Fe, Mn, Ni and Pb in water samples from the study area and the reference area were 0.006 ± 0.002 and 0.001 ± 0.000; 0.002 ± 0.001 and 0.001 ± 0.001; 0.012 ± 0.001 and 0.018 ± 0.015; 1.001 ± 0.502 and 0.610 ± 0.165; 0.270 ± 0.138 and 0.012 ± 0.003; 0.320 ± 0.199 and 0.010 ± 0.002; and 0.019 ± 0.006 and 0.023 ± 0.019 mg/L, respectively. Statistical analyses results indicated that there were significantly different (*p* < 0.05) among the concentrations of As, Mn and Ni between the study and the reference areas. Furthermore, the results showed that Fe, Mn and Ni in water samples exceeded the standard values (Table 3).

### 3.3. Heavy Metal and Metalloid Concentrations in Sediment Samples

The average concentrations of As, Cd, Cr, Fe, Mn, Ni and Pb in sediment samples between the study and the reference areas were 6.78 ± 1.86 and 0.03 ± 0.02; 2.45 ± 0.58 and 0.073 ± 0.06; 20.71 ± 6.67 and 0.48 ± 0.22; 8219.58 ± 7574.09 and 214.94 ± 59.49; 301.34 ± 140.31 and 2.95 ± 1.06; 36.50 ± 6.29 and 1.89 ± 1.79; and 68.71 ± 19.02 and 1.44 ± 0.93 mg/kg, respectively. The statistical analyses results indicated that there were significantly different of all studied heavy metal and metalloid concentrations in sediment samples between the study area and the reference area (*p* < 0.05). Furthermore, the results showed that As and Cd in sediment samples exceeded the standard values (Table 4).

### 3.4. Heavy Metal and Metalloid Concentrations in O. vittatus Muscles

The average concentrations of As, Cd, Cr, Fe, Mn and Ni in *O. vittatus* muscle samples from the study area and from reference area are 3.48 ± 3.38 and 0.37 ± 0.44; 0.05 ± 0.01 and 0.02 ± 0.00; 4.72 ± 1.76 and 1.67 ± 0.13; 56.10 ± 18.30 and 39.65 ± 5.10; 14.76 ± 4.22 and 0.56 ± 0.07; and 1.76 ± 0.45 and 0.62 ± 0.19 mg/kg, respectively, whereas the concentrations of Pb from both areas were undetected (Table 5). As, Cd, Cr, Mn and Ni concentrations accumulated in the fish muscles from both areas were significantly different (*p* < 0.05), whereas the Fe concentrations in the fish muscles from both areas were not significantly different (*p* > 0.05). The standard concentrations of Fe and Ni in the fish muscles are not officially declared.

### 3.5. Chromosome Assessment in O.vittatus

The diploid chromosome number (2n) of *O. vittatus* from the study area and the reference area were 2n = 50. The normal karyotype of *O. vittatus* from both areas consist of 8 pairs of metacentric, 10 pairs of sub-metacentric and 7 pairs of acrocentric regions (Figure 2). Figure 3 shows the different types of chromosome aberrations in the metaphase kidney cells. The results found seven types of chromosome aberrations from *O. vittatus* in the Sepon gold-copper mine area as fragmentation (F), single chromatid gap (SCG), iso-chromatid gap (ISCG), centric gap (CG), deletion (D), single chromatid break (SCB) and centric fragmentation (CF) with number of chromosome aberrations as 13, 41, 9, 253, 53, 5 and 51, respectively. Total number of chromosome aberrations, cell number with chromosome aberrations and percentage of chromosome aberrations of *O. vittatus* in the study and the reference areas were 425 and 63, 96 and 37, and 38.40 and 14.80, respectively. The statistical analyses indicated that total number of chromosome aberrations, cell number with chromosome aberrations and percentages of chromosome aberrations from both areas were significantly different (*p* = 0.042 and 0.043) (Table 6). 

### 3.6. Serum Biochemistry

The average and standard deviation values of serum total protein, glucose and liver enzyme parameters of *O. vittatus* show in Table 7. The total protein and liver enzymes (AST and ALT) of the fish from the study area near the Sepon gold-copper mine were significant and higher than of the fish in the reference area (*p* < 0.05).

### 3.7. Histopathology

Hepatocytes of *O. vittatus* from the study area near the Sepon gold-copper mine revealed pathologic structures at nuclei and cytoplasmic organelles as nuclear membrane degeneration, rough endoplasmic reticulum disintegration and abnormal cytoplasmic mitochondria (Figure 4).

## 4. Discussion

### 4.1. Water Quality Parameters

All measured water quality parameters were within standards of uncontaminated surface water [46] and suggested that the water conditions from the reference area were suitable for fish to live, especially temperature, dissolved oxygen, pH and electrical conductivity [47,48,49,50,51].

### 4.2. Heavy Metal and Metalloid Concentrations in Water, Sediment and O. vittatus

The concentrations of As, Mn and Ni in water samples from the study area near the Sepon gold-copper mine and the reference area were significantly different (*p* < 0.05), whereas the concentrations of Cd, Cr, Fe and Pb in the water from the study and the reference areas were not significantly different (*p* > 0.05) (Table 3). However, the concentrations of Fe, Mn and Ni in the water from the study area exceeded the limits of water standards [36,37,38]. The results were in accordance with other studies. Intamat et al. [52] reported that concentrations of As, Cd, Fe and Mn in water reservoirs around gold mine area in Loei Province of Thailand were higher but not significantly different compared with the reference site. Jiang et al. [53] detected that concentration of Cr in water from Lake Caizi in Southeast China was higher than of the non-contaminated area. Furthermore, concentrations of As and Mn from other studies in Thailand, such as in the Loei gold mine, Thailand community reservoirs and Mahasarakhm domestic wastewater canal were higher than the results in this study [50,54,55,56].

The concentrations of heavy metals and metalloids in sediment from both studied areas indicated that they were statistically significant different (*p* < 0.05). The average concentrations of As and Cd in sediments samples of the Nam Kok river near the Sepon gold-copper mine were higher than of the standard values [39,40], while the averages concentration of Cr, Fe, Mn, Ni and Pb in sediment were lower than the standard values. These results suggested that As and Cd as major accumulated metals in the sediment of the Nam Kok river were relatively high, as reported in other research studies. Culioli et al. [50] found a significantly high concentration of As in a river of Corsica under an influence of tailing way of the gold mine, and Intamat et al. [52] and Tengjaroenkul et al. [57] found that As and Cd were the two highest concentrations among the sediment metals around gold mine area of Loei Province, Thailand. Furthermore, Intamat et al. [52], Khamlerd et al. [56] and Phoonaploy et al. [58] reported that concentrations of Cd, Cr, Pb, Ni, Zn and Mn were lower than of the metals reported in this study. In general, most of human activities related to mining, electrical appliances, jewellery and agriculture can release As, Cd, Cr, Fe, Mn, Ni and Pb into water, and later accumulated in the sediment of the surrounding environment. The toxic metals, particularly at high levels can be absorbed and directly induce detrimental effects to aquatic animals as well as cause negative signs and symptoms to human [59] who situate at the top of the food chain and ecosystem.

The average concentrations of heavy metals and metalloids in the *O. vittatus* muscles are summarized in Table 5. Besides Fe, the concentrations of other heavy metals and metalloid in fish muscle samples from the study and the reference areas were statistically significant (*p* < 0.05). The concentrations of As, Cd, Cr and Mn in *O. vittatus* muscles from the study area were higher than the standard values for freshwater fish [36,42,43,45]. Furthermore, the contaminated heavy metals and metalloids in sediment and water samples in this study (Table 3 and Table 4) can be absorbed into algae and aquatic plants via root resulting the accumulation mainly in stem and leaves, and finally the metals and metalloid can be bioaccumulated in the herbivorous fish including *O. vittatus* after being fed via the digestive tract, besides being directly absorbed from aquatic polluted ecosystem into their gills and skin. Jiang et al. [53] reported that concentrations of Cd and Cr in the muscles of *Cyprinus carpio* and *C. auratus* from Lake Caizi in Southeast China were higher than of those in the Nam Kok river. Keshavarzi et al. [60] demonstrated that concentrations of As, Cd and Pb in fish species (*Anodontostoma chacunda* and *Cynogloddurs arel*) from Persian Gulf were higher than of the metals in *O. vittatus* muscles. Intamat et al. [52] found As, Cr, Cd, Ni, Fe and Mn in muscles of *Rasbora tornieri* around the gold mine area of Loei province, Thailand. Neeratanaphan et al. [55] reported the concentrations of As, Cd and Cr in *Esomus metallicus*. Keshavarzi et al. [60] detected As and Cd in *Anodontostoma chacunda*, while Khamlerd et al. [56] found concentrations of As, Cd, Cr and Ni in *Channa striata* at Bueng Jode reservoir, Khon Kaen Province, Thailand.

### 4.3. Chromosome Study in O.vittatus 

This study revealed seven types of chromosome aberrations, including F, SCG, ISCG, CG, D, SCB and CF with the number of chromosome aberrations as 13, 41, 9, 253, 53, 5 and 51, respectively. For the chromosome aberrations, the statistical analyses indicated that total number of chromosome aberrations, number of cell with chromosome aberrations and percentages of chromosome aberrations between the fish samples from the study area near the Sepon gold-copper mine and the reference area were significantly different (*p* = 0.042 and 0.043). Structural abnormalities of chromosomes in *O. vittatus* in this study revealed as reported in several previous studies. Neeratanaphan et al. [55] demonstrated six chromosome aberrations, including CF, CG, SCG, F, D and P in *Esomus metallicus* collected from a gold mine area in the Wang Saphung District of Loei Province, Thailand. Tengjaroenkul et al. [61] showed in the laboratory scale that five types of chromosome aberrations, including SCG, SCB, CG, F and D in *O. vittatus* exposed with sodium As at different concentrations. Khamlerd et al. [56] revealed seven types of chromosome aberrations, including SCG, SCB, CF, CG, F, D and SCD in *Channa striata* contaminated with As, Pb, Cd, Cr, Cu, Fe, Zn, Mn and Ni from Bueng Jode reservoir, Khon Kaen Province, Thailand. In addition, Neeratanaphan et al. [62] found seven types of chromosome aberrations, including SCG, SSCG, SCB, CF, F, SSCF and D in Asian swamp eel (*Monopterus albus*) followed by the As contamination near a gold mine area, Loei Province, Thailannd. These above findings revealed that chromosome aberrations could be influenced by fish species, habitats and kinds of exposed pollutants, etc.

Scientists have proposed to explain the genotoxicity of several heavy metals, especially Cd, Ni, As and Pb affected to aquatic animals. Among heavy metals, Cd, Ni, As can induce point mutation, deletion, ploidy change, substituted bases, gene amplification, genetic expression, cell proliferation, interference in DNA methylation, DNA damage, DNA repair inhibition and histone modification [63,64,65,66,67,68,69]. Pb can cause several indirect genotoxicity mechanisms, mitogenesis, DNA damage and alterations in gene transcription [69].

### 4.4. Serum Biohemistry Study in O.vittatus

Serum TP, Glu and liver enzymes (AST and ALT) of the fish from the study area near the Sepon gold-copper mine were significantly higher than of the fish in the reference area (*p* < 0.05). Their levels are important in monitoring fish health as well as the aquatic ecosystem. TP measurements are used in the diagnosis and treatment of a variety of diseases involving the liver, kidney and bone marrow as well as other metabolic and nutritional disorders. Generally, serum protein is responsible for transportation, accumulation and detoxification of metals in fish that usually associates with toxic substances and stress due to environmental changes [70]. In this study, TP of the gold-copper mine fish were significantly different and higher than of the reference fish. Therefore, the increases of serum protein levels, particularly after exposure to hazardous metals in *O. vittatus*, indicated that the organs were repairing their damages [71]. A similar finding was revealed in *Cyprinus carpio* with Pb and Cd contaminations [70]. For serum Glu, the levels in *O. vittatus* were not significantly different between the Sepon gold-copper mine and the reference fish, and were similar to other reports [70,71,72] which treated Pb and Cd in *C. carpio*. For ALT and AST, the enzyme measurements are valuable in the diagnosis of hepatic disorders. In this study, AST and ALT levels demonstrated significant differences between the fish near the Sepon gold-copper mine and the reference areas (*p* < 0.05), and suggested an occurrence of hepatic injury or liver impairment likely due to metal accumulation in this organ of *O. vittatus* as in other fish [31,73]. Rao et al. [74] found an increase in the level of AST in the maternal tissue of *H. fulvipes* as a response to the stress induced by the heavy metals. Naga et al. [75] demonstrated significant consequences in plasma enzymes (AST and ALT) after exposure to Cd in marine fish *Mugil seheli*. Asgah et al. [73] reported that Cd increased AST and ALT in the Nile tilapia *O. niloticus* as concentration and exposure time dependent manner. Furthermore, heavy metals can negatively affect fish health in terms of protein, carbohydrate and lipid profiles [71].

### 4.5. Histopathology Study in O.vittatus

Histopathology study is an important discipline to evaluate morphology changes in cells and tissues. The liver is the vital organ for detoxification and accumulation of toxic substances. Heavy metals adversely affect the functions of different aquatic animal organs, especially changes of hepatic enzyme activities, extravasation of blood and necrosis of the liver cells, fusion of gill lamellae, and genotoxicity [76,77,78,79]. In this study, the hazardous metals in *O. vittatus* accumulated in the muscle affected not only the fish chromosomes, but also the liver cell structures. Generally, heavy metal toxicity depends on the organ of exposure, fish species, time of exposure, concentrations of toxicants and metabolic processes in the organs of each organism [31,60,73,80].

The abnormal liver cell structures in *O. vittatus* included nuclear membrane degeneration, rough endoplasmic reticulum disintegration and abnormal cytoplasmic mitochondria. These changes suggested that the heavy metal concentrations in *O. vittatus* near the Sepon gold-copper mine could induce the morphological changes of the liver cells. The results were in accordance with several previous studies. Giari et al. [32] found mitochondria and endoplasmic reticulum defects after exposure to Cd. Dyk et al. [80] found that the Cd exposure could damage the liver cell structures of *Oreochromis mossambicus*. Mishra and Mohanty [81] revealed that the occurrences of degenerative nuclei could be induced by metal free radicals. Vinodhini and Narayanan [82] reported that the accumulated heavy metals could cause cellular degeneration in the liver of *Cyprinus carpio*.

Heavy metals and metalloid accumulation in *O. vittatus* affecting chromosomes, serum biochemistry and histopathology could lead to a decrease in the fish population and aquatic animals near Sepon gold-copper mine area. The *O. vittatus* is a major fish food for people in Lao People’s Democratic Republic besides the people who inhabit along the Nam Kok river near the Sepon gold-copper mine. Furthermore, as the fish are situated at the top level of the food chain, human tends to accumulate metal and metalloid elements at toxic levels in their bodies after consuming the metal contaminated *O. vittatus* on a daily basis for a length of time.

## 5. Conclusions

The concentrations of heavy metals and metalloids (As, Cd, Cr, Fe, Mn, Ni and Pb) in water, sediment and fish muscle samples in the study area were higher than of those from the reference area. The As, Cd, Cr and Pb concentrations in sediment were higher than the water samples from the study area, and did not exceed the Thailand and WHO standards, while the averages of As, Cd, Cr and Mn concentrations in *O. vittatus* from the study area exceeded the food quality standards of the EC and WHO. Furthermore, these heavy metals and metalloids affected *O. vittatus* fish in terms of chromosomal aberration, serum biochemical changes and the liver cell structure alterations. Results from this study will provide the basis for monitoring the environment surrounding the mining area, and contribute to efforts to improve metal contamination involving the food safety for people near the contaminated area and for environmental management.

## Figures and Tables

**Figure 1 ijerph-17-09492-f001:**
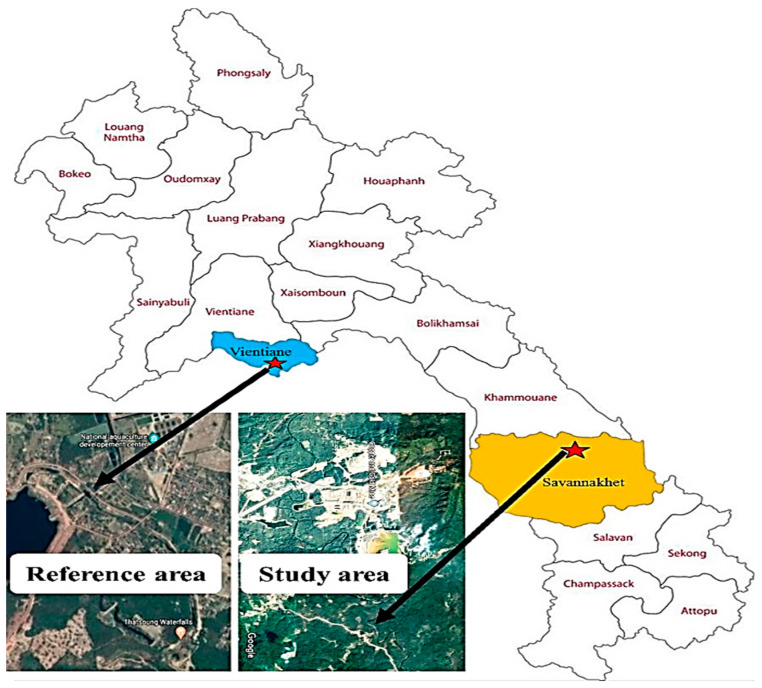
Geographical maps of the Nam Souang river as the reference area (**Left**) and the Nam Kok river near the Sepon gold-copper mine as the study area (**Right**).

**Figure 2 ijerph-17-09492-f002:**
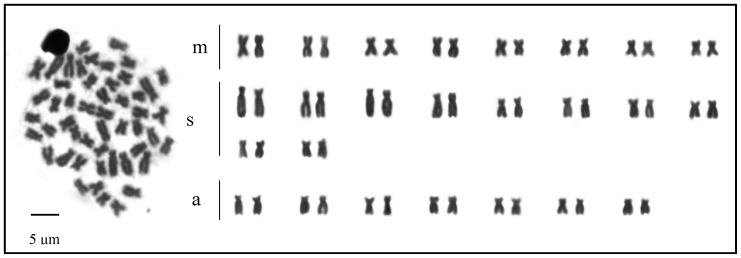
Karyotype of normal chromosome of *O. vittatus* (2n = 50). Remarks: metacentric (m), sub-metacentric (sm), acrocentric (a).

**Figure 3 ijerph-17-09492-f003:**
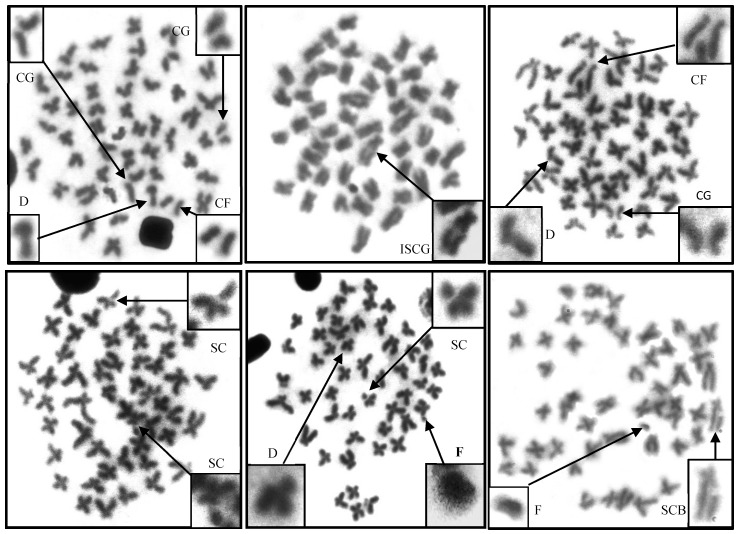
Seven types of chromosome aberrations in the metaphase spread cells of *O. vittatus*. Remarks: centromere gap (CG), deletion (D), centric fermentation (CF), isochromatid gap (ISCG), single chromatid gap (SCG), fragmentation (F), single chromatid break (SCB).

**Figure 4 ijerph-17-09492-f004:**
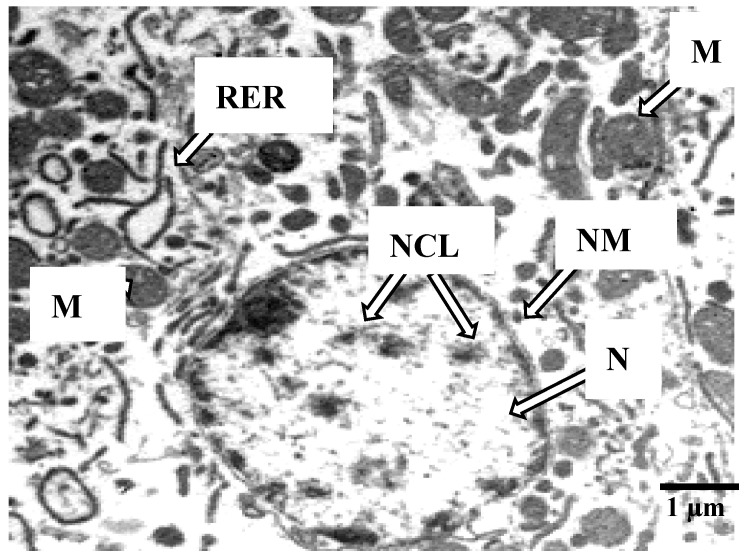
Electron micrograph of *O. vittatus* hepatocyte. Remarks: Nucleus (N), Nucleolus (NCL), Nuclear membrane (NM), Mitochondria (M), Rough endoplasmic reticulum (RER); Bar = 1 μm.

**Table 1 ijerph-17-09492-t001:** Analytical methods used for measurements of water quality parameters.

Water Quality Parameters	Analytical Methods
Dissolved oxygen	DO meter, Model 966, Mettler Toledo
pH	pH meter, Model EcoScan pH 5
Temperature	Eutech Thermometer
Total hardness, carbonate hardness	Test kits, Chulalongkorn University, Thailand
Electrical conductivity	EC meter, Mettler Toledo
Nitrite, Nitrate, Ammonia	Titration methods

**Table 2 ijerph-17-09492-t002:** The water quality parameters of the Nam Kok river near the Sepon gold-copper mine as the study area and the Nam Souang river as the reference area.

Samples	Concentrations		*p*-Value
	Nam Kok River (Study Area)	Nam Souang River (Reference Area)	
Temperature (°C)	24.41 ± 0.36	25.76 ± 1.35	0.005 *
DO (mg/L)	6.80 ± 0.61	7.57 ± 0.57	0.006 *
pH	8.02 ± 0.25	7.68 ± 0.47	0.005 *
TH (ppm)	9.10 ± 0.48	3.53 ± 0.46	0.108
CH (ppm)	108.00 ± 16.43	98.67 ± 5.93	0.246
Nitrite (mg/L)	0.031 ± 0.003	0.045 ± 0.005	0.126
Nitrate (mg/L)	7.57 ± 2.24	8.70 ± 3.47	0.065
Ammonia (mg/L)	0.02 ± 0.01	0.09 ± 0.02	0.051
EC (μs/cm)	427.00 ± 14.12	246.60 ± 8.41	0.193

* Pooled standard error (*p*-value) of statistically significant difference (*p* < 0.05). Remarks: dissolved oxygen (DO), potential of hydrogen (pH), total hardness (TH), carbonate hardness (CH) and electrical conductivity (EC); (mean and standard deviation; *n* = 3).

**Table 3 ijerph-17-09492-t003:** Heavy metal and metalloid concentrations in water samples (n = 10) of the Nam Kok river near the Sepon gold-copper mine as the study area and the Nam Souang river as the reference area.

Heavy Metals	Concentration in Water (mg/L)		*p*-Value	Standard
	Nam Kok River (Study Area)	Nam Souang River (Reference Area)		
As	0.006 ± 0.002	0.001 ± 0.000	0.008 **	0.01 ^a^
Cd	0.002 ± 0.001	0.001 ± 0.001	0.209	0.01 ^b,c^
Cr	0.012 ± 0.001	0.018 ± 0.015	0.602	0.05 ^b,d^
Fe	1.001 ± 0.502 *	0.610 ± 0.165 *	0.617	0.3 ^e^
Mn	0.270 ± 0.138 *	0.012 ± 0.003	0.009 **	0.05 ^f^
Ni	0.320 ± 0.199 *	0.01 ± 0.002	0.009 **	0.02 ^d^
Pb	0.019 ± 0.006	0.023 ± 0.019	0.754	0.05 ^b^

* Metal or metalloid concentration exceeds the standard reference value. ** Pooled standard error (*p*-value) of statistically significant difference (*p* < 0.05). Water standard references ^a^ [33], ^b^ [34], ^c^ [35], ^d^ [36], ^e^ [37], ^f^ [38].

**Table 4 ijerph-17-09492-t004:** Heavy metal and metalloid concentrations in sediment samples (n = 10) from in the Nam Kok river near the Sepon gold-copper mine as the study area and the Nam Souang river as reference area.

Heavy Metals	Concentration in Sediment (mg/kg)		*p*-Value	Standard
	Nam Kok River (Study Area)	Nam Souang River (Reference Area)		
As	6.78 ± 1.86 *	0.03 ± 0.02	0.009 **	3.90 ^a^
Cd	2.45 ± 0.58 *	0.073 ± 0.06	0.009 **	0.16 ^b^
Cr	20.71 ± 6.67	0.48 ± 0.22	0.009 **	45.50 ^b^
Fe	8219.58 ± 7574.09	214.94 ± 59.49	0.009 **	-
Mn	301.34 ± 140.31	2.95 ± 1.06	0.009 **	1800 ^a^
Ni	36.50 ± 6.29	1.89 ± 1.79	0.010 **	75 ^c^
Pb	68.71 ± 19.02	1.44 ± 0.93	0.009 **	400 ^a^

* Metal or metalloid concentration exceeds the standard reference value. ** Pooled standard error (*p*-value) of statistically significant difference (*p* < 0.05). Sediment standard references ^a^ [39], ^b^ [40], ^c^ [41].

**Table 5 ijerph-17-09492-t005:** The heavy metal and metalloid concentrations in *O. vittatus* samples (n = 10) from the Nam Kok river near the Sepon gold-copper mine as the study area and the Nam Souang river as the reference area.

Heavy Metals	Concentration (mg/kg)		*p*-Value	Standard
	Nam Kok River (Study Area)	Nam Souang River (Reference Area)		
As	3.48 ± 3.38 *	0.37 ± 0.44	0.047 **	2.0 ^a^
Cd	0.05 ± 0.01 *	0.02 ± 0.001	0.009 **	0.05 ^b^
Cr	4.72 ± 1.76 *	1.67 ± 0.13	0.009 **	2.0 ^c^
Fe	56.10 ± 18.30	39.65 ± 5.10	0.347	-
Mn	14.76 ± 4.22 *	0.56 ± 0.07	0.016 **	1.0 ^c^
Ni	1.76 ± 0.45	0.62 ± 0.19	0.009 **	-
Pb	ND	ND	-	0.2 ^d^

* Metal or metalloid concentration exceeds the standard reference value. ** Pooled standard error (*p*-value) of statistically significant difference (*p* < 0.05). Muscle Standard references ^a^ [42], ^b^ [43], ^c^ [44], ^d^ [45].

**Table 6 ijerph-17-09492-t006:** Total number of chromosome aberrations, cell number with chromosome aberrations and percentage of chromosome aberrations of *O. vittatus* from the Nam Kok river near the Sepon gold-copper mine as the study area and the Nam Souang river as the reference area (median (interquartile range), n = 5).

*O. vittatus*	Number of Chromosome Aberrations							Total Number of CA	Cell Number with CA	Percentage of CA
	F	SCG	ISCG	CG	D	SCB	CF			
Study area	2 (2)	10 (5.5)	1 (3.5)	50 (20.5)	10 (1.5)	1 (2)	9 (7)	82 (11.5)	19 (3.5)	38 (7)
Total/Average *	13	41	9	253	53	5	51	425	96	38.40 *
Reference area	0 (0.5)	0 (1)	0 (0.5)	9 (3.5)	2 (1.5)	0 (0)	1 (1.5)	13 (4)	8 (2.5)	16 (5)
Total/Average *	1	2	1	44	11	0	4	63	37	14.80 *
*p*-value								0.043 **	0.042 **	0.042 **

***** Average percentage of chromosome aberrations; ****** Statistically significant difference (*p*-value < 0.05). Remarks: F = fragmentation, SCG = single chromatid gap, ISCG = isochromatid gap, CG = centromere gap, D = deletion, SCB = single chromatid break, CF = centric fragment, CA = chromosome aberrations.

**Table 7 ijerph-17-09492-t007:** Serum enzyme parameters of *O. vittatus* from the Nam Kok river near the Sepon gold-copper mine as the study area and the Nam Souang river as the reference area.

Parameter	Nam Kok River (Study Area; *n* = 3)	Nam Souang River (Reference Area; *n* = 3)	*p*-Value
TP (mg/mL)	4.82 ± 0.72 *	3.25 ± 0.34 *	0.026
Glu (mg/dL)	37.23 ± 4.01	43.27 ± 4.06	0.121
AST (IU/L)	58.40 ± 12.34 *	25.67 ± 3.74 *	0.012
ALT (IU/L)	64.00 ± 9.67 *	22.67 ± 6.87 *	0.004

* significantly different in the same row (*p* < 0.05). Remarks: Total protein (TP), Glucose (Glu), Aspartate aminotransferase (AST), Alanine aminotransferase (ALT).

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
