# Peer review of "Effects of Contamination by Heavy Metals and Metalloids on Chromosomes, Serum Biochemistry and Histopathology of the Bonylip Barb Fish Near Sepon Gold-Copper Mine, Lao PDR"

_ijerph, 2020, doi:10.3390/ijerph17249492_

Round 1
Reviewer 1 Report
This manuscript not only monitored the contents of metals and metalloids in the water and sediments nearby the gold-copper mine, but also studied the accumulation of metals and metalloids in the fish and its possible adverse effects on the pathological mechanism.
The authors found that the toxic effects of metals and metalloids on the fish in the river nearby the mining areas, such as chromosomal aberration, serum biochemical changes and the liver cell structure alterations. The results are very interesting and valuable.
In general, the data and results are reliable. English expression is basically fluent, but should be polished further.
Line 20: CA??
Line 33: concession to foreign companies??
Line 40: It should be “Besides mining industry,”
Line 43: in some polluted area
Line 63: like other countries in Southeast Asia
Line 69: what does “CA” mean?
Line 96: how do you study “liver cell structure changes”
Line 269-271: this paragraph should be deleted. As the water of the study area near the gold-copper mine is significantly high in some heavy metals, how can you say the water condition was suitable for fish?
Line 398: provide
Line 393: The metal content in the sediment was certainly higher than that in the water. You should compare the contents of heavy metals in the river sediments nearby the mining areas and the control.
Author Response
|
Reviewer 1 |
Explanation in revision manuscript |
|
1) Line 20: CA?? |
CA= “chromosome aberrations” Edited in the yellow highlight text. (whole manuscript ) |
|
2) Line 33: concession to foreign companies?? |
Line 34: concession to foreign companies (correct in green highlight) |
|
3) Line 40: It should be “Besides mining industry,” |
Edited in the yellow highlight text. (Line 41-42) |
|
4) Line 43: in some polluted area |
Edited in the yellow highlight text. (Line 44) |
|
5) Line 63: like other countries in Southeast Asia |
Edited in the yellow highlight text. (Line 64) |
|
6) Line 69: what does “CA” mean? |
CA= “chromosome aberrations” Edited in the yellow highlight text. |
|
7) Line 96: how do you study “liver cell structure changes” |
Edited in the yellow highlight text. The author edited as “Liver cell structure changes were investigated by compare the liver cells between the fish from the reference and the study areas.” (Line 97-99) |
|
8) Line 269-271: this paragraph should be deleted. As the water of the study area near the gold-copper mine is significantly high in some heavy metals, how can you say the water condition was suitable for fish? |
Edited in the yellow highlight text. (Deleted “the study area”) (Line 280-282) |
|
9) Line 398: provide |
Edited in the yellow highlight text. (Line 409) |
|
10) Line 393: The metal content in the sediment was certainly higher than that in the water. You should compare the contents of heavy metals in the river sediments nearby the mining areas and the control. |
Edited in the yellow highlight text. “The As, Cd, Cr, and Pb concentrations in sediment were higher than the water samples from the study area” (Line 404-405) |

Reviewer 2 Report
Manuscript Number: ijerph-1034268
Title: Effects of Contaminated Heavy Metals and Metalloid on Chromosome, Serum Biochemistry and Histopathology of the Bonylip Barb Fish Near Sepon Gold-Copper Mine, Lao PDR
Authors: Latsamy Soulivongsa, Bundit Tengjaroenkul, Lamyai Neeratanaphan
[Major comments]
This study has well summarized and provide useful information on metal contamination in Laos. I recommend this manuscript is published in the journal, but there are several points to be revised for the objective.
Critical issue in the manuscript is authors have mistakes about concentration value and statistical result. You have performed Mann Whitney U-test, but you have shown “mean” as representative data. For this statistical result, authors should use “median”. Moreover, before the data treatment, data distribution should be checked and then suitable statistical analysis followed by suitable value should be used. I recommend authors perform such approach.
This study investigation has conducted in Laos. Is there no coworkers (coauthors) in Laos side?
[Minor comments]
Title
“Contaminated heavy metals” is somehow wrong. “Contamination by heavy metals” is correct, I think.
Sample collection
There is no description when you have collected samples.
Table 6
No need individual values.
Table 7
Style should be same as tables 1-3. I mean you add p-value column.
Reference
Reference number is overlapped, there is two same numbering in each reference.
Author Response
|
Reviewer 2 |
Explanation |
|
1) Critical issue in the manuscript is authors have mistakes about concentration value and statistical result. You have performed Mann Whitney U-test, but you have shown “mean” as representative data. For this statistical result, authors should use “median”. Moreover, before the data treatment, data distribution should be checked and then suitable statistical analysis followed by suitable value should be used. I recommend authors perform such approach. |
Edited in the yellow highlight text. (Line 153-159) |
|
2) This study investigation has conducted in Laos. Is there no coworkers (coauthors) in Laos side? |
The first author work at the University from Laos PDR as a lecturer, but now he is PhD student in Khon Kaen University, Thailand. |
|
3) Title “Contaminated heavy metals” is somehow wrong. “Contamination by heavy metals” is correct, I think.
|
Edited in the yellow highlight text. |
|
4) Sample collection There is no description when you have collected samples. |
Edited (Line 93) |
|
5) Table 6 No need individual values. |
Edited in the yellow highlight text. |
|
6) Table 7 Style should be same as tables 1-3. I mean you add p-value column. |
Edited in the yellow highlight text. |
|
7) Reference Reference number is overlapped, there is two same numbering in each reference. |
Edited |
